# Psychometric Properties and Validation of the Italian Version of Ages & Stages Questionnaires Third Edition

**DOI:** 10.3390/ijerph20065014

**Published:** 2023-03-12

**Authors:** Filippo Manti, Federica Giovannone, Matteo Ciancaleoni, Gloria De Vita, Francesca Fioriello, Federica Gigliotti, Carla Sogos

**Affiliations:** 1Unit of Child Neurology and Psychiatry, Department of Human Neuroscience, Sapienza University of Rome, 00185 Roma, Italy; 2Hogrefe Editore, 50132 Firenze, Italy

**Keywords:** ASQ-3, Psychometric properties, Italian adaptation, developmental delay, screening tool

## Abstract

**Objectives**: The Ages & Stages Questionnaires Third Version (ASQ-3) identifies the risk of developmental delay in children aged 1 to 66 months. The aim of this study was to determine a reliable and valid instrument for the Italian population to enable the screening of children’s development. **Methods**: Data from 2278 Italian children (age range: 1–66 months) were used to evaluate item discrimination power using the corrected item-total correlation. Internal consistency was analyzed by Cronbach’s alpha scores and a Confirmative Factor Analysis was conducted to test the factor structure of the test. Data were also collected to examine the ASQ-3 test-retest reliability and concurrent validity, which was investigated using the Griffiths Scales of Child Development, Third Edition, the Peabody Developmental Motor Scale, Second Edition, and the Developmental Profile, Third Edition tools. In order to evaluate discriminant validity, differences between typical development children and several clinical groups have been performed. Finally, two different cut-off scores have been proposed. **Results**: The results showed that the questionnaires are composed of high-quality items; the original factor structure has been confirmed and strong Pearson product-moment correlation coefficients between the overall and the total for each domain (ranging from 0.73 to 0.88). The Italian version of the ASQ-3 had adequate internal consistency and a strong agreement between observations with two weeks’ intervals. Moreover, the test showed a high discriminant validity due to the possibility of fully discriminating between typical development children and several clinical groups. Finally, two different cut-off scores have been identified using ROC curves in order to have a screening and a diagnostic cut-off value. **Conclusion**: This study evaluated the psychometric properties of the Italian adaptation of ASQ-3 questionnaires. We demonstrated the validity of the ASQ-3 and determined new cut-off scores for Italian children. Early identification and accurate assessment are important starting points to better understand and anticipate the needs of children and their link to services.

## 1. Introduction

Neurodevelopmental disorders (NDs) encompass a highly heterogeneous group of diseases characterized by impairments in communication, cognition, behavior, and motor functioning. These disorders are likely to result from a combination of genetic, biological, psychosocial, and environmental risk factors [1]. The prevalence of NDs is estimated to be between 5% and 15%, with 1% to 3% having an overall global developmental delay or an intellectual disability.

In the USA several epidemiological studies have reported that 12–16% of children suffer from NDs [2,3,4]. By contrast, other studies have reported that between 4 and 10% of children in the general population have NDs, although only 30% of these are identified before primary school [5,6]. However, it is well-known that without the use of standardized instruments, no more than 30% of children with developmental delays can be identified, potentially delaying access to early intervention services [7,8]. Early identification of NDs is crucial to improve the quality of life of children and their families and provide appropriate therapies and education [9]. There is evidence suggesting that early detection of NDs allows for timely, effective intervention [10,11].

For these reasons, developmental screening tools are now being increasingly deployed and can be administered directly (e.g., Denver Developmental Screening Tool-II) or reported by parents (e.g., Parents’ Evaluation of Developmental Status) [12,13].

The American Academy of Pediatrics (AAP) recommends the application of standardized developmental screening tools during the regular well-child visits of infants and preschool children at different ages (9, 18, 24, and 30–36 months), especially when certain well-known risk factors are at play (e.g., premature birth and genetic or metabolic problems) [14,15]. It is important to implement valid screening tools, not only for children with suspicions of NDs but also for children having typical development, as a way to increase families’ awareness of child development aspects and expected behavior [16].

The ASQ-3 is a developmental screening tool widely used by clinicians, researchers, and intervention programs across numerous countries [17,18,19,20,21,22,23,24,25,26,27].

Several reports indicate that the ASQ-3 has well-established psychometric properties in a clinical context, such as test-retest reliability, internal consistency, criterion validity, sensitivity, and specificity [16,17,18,19,28]. The ASQ-3 validity tested on 18,000 questionnaires administered in the United States yielded an overall agreement across questionnaires of 86%, with a range of 73%–100%. Sensitivity (i.e., children in whom the ASQ detected a developmental delay and who had a delay according to standardized assessment) ranged from 85% to 92%, whereas specificity (i.e., children in whom the ASQ-3 did not detect a delay and whose development was normal according to the standardized assessment) ranged from 78%–92%. Validity and reliability ranged from 70% to 100% [16,17,18,19]. It has subsequently been standardized and validated for cross-cultural use in many parts of the world. Some studies found relatively high and consistent validity of the ASQ-3 for all age groups. Romero Otalvaro et al. [23] observed that the ASQ-3 met the psychometric properties necessary for a targeted and systematic assessment of development in Argentine children. Van Heerden et al. [29] administered the ASQ-3 to 853 children living in South Africa and Zambia for identifying at-risk children in low- and middle-income countries. Strong psychometric properties of the Spanish ASQ-3 have also been demonstrated when it is used with older children and children at high-risk or with more severe delays [30].

In the present study, the questionnaires were translated from English into Italian. Changes were made in the syntactic structure, which was characterized by removing particles or elements with redundant meanings, and culturally adapting some terms since they were not commonly used in Italian. The final version was therefore included in the study.

The aim of this study was to determine a reliable and valid instrument for the Italian population to enable the screening of children’s development. This study was carried out for the purpose of cultural adaptation, validation, and standardization of the ASQ-3 questionnaire for 1–66-month-old children living in Italy. We expected to demonstrate that the ASQ-3 questionnaire displays all the characteristics making it a useful and psychometrically adequate tool in the identification of NDs.

## 2. Methods

### 2.1. Adaptation Procedure

The schools’ principals and the teachers’ coordinators were contacted by phone, email, or in person to verify if they were interested in taking part in the research. The objectives of the study, including procedures as well as the aims and the methodology, were explained in advance of the distribution of the questionnaires. All the principals agreed to the project and consequently allowed us to contact the children’s parents through the teachers’ coordinators, who ensured that the questionnaire was sent to and returned by the parents via email (the information was collected in electronic format). Informed consent was obtained from the parents of children who agreed to participate in the study. All the parents were informed that the information collected would be confidential; the questionnaires were anonymous; the participation in the study was exclusively on a voluntary basis, and explaining the importance of early identification of developmental delays was used to increase the participant response rate. Twenty-one incomplete questionnaires were eliminated. The parents of the children were then asked to complete the Ages and Stages Questionnaire-3 (see Measures). All the data were collected in the years 2020–2021.

The participating centers are geographically spread across Italy including both rural and urban areas.

The parents of children who fell within the clinical range in one, or more than one, domain of the ASQ-3 were contacted by email or telephone by a clinician of the neuropsychiatric services, who suggested that a meeting be arranged to discuss the ASQ-3 data. This first observation was followed, if deemed necessary, by a comprehensive clinical assessment (language, cognitive, motor, and adaptive domains) of the child, during which the disorder was either confirmed or not confirmed (the assessment took about 3–4 h). Children were thus offered the opportunity to promptly start targeted treatment directly involving both parents and schools.

The study was approved by the local Ethics Committee of Sapienza University of Rome. The protocol for the research project conformed to the provisions of the Declaration of Helsinki.

### 2.2. Participants

A total of 2278 Italian children (M = 1238; F = 1040; age range 1–66 months) took part in the study. 1874 children (M = 987; F = 887; age range 1–66 months) were recruited from kindergartens located in Italy. 404 children (M = 251; F = 153; age range 1–66 months) with NDs were recruited from the Department of Human Neuroscience, Sapienza University of Rome.

### 2.3. Analytical Procedures

The internal consistency of ASQ-3 items was examined using correlational analyses and the Cronbach coefficient alpha [31]. Item means and standard deviations were calculated, as well as the item-total correlation.

Test–retest reliability was analyzed by comparing ASQ-3 questionnaires completed by the same parent within a two-weeks interval. When completing the first questionnaire, the parents did not know that they would be asked to complete a second questionnaire.

### 2.4. Measures

The ASQ-3 is a screening tool used to assess development during the first five years of life [28], and is composed of 21 questionnaires to be used on infants (2, 4, 6, and 8 months), toddlers (9, 10, 12, 14, 16, 18, 20, 22, 24, 27, 30, and 33 months) and preschoolers (36, 42, 48, 54, and 60 months). The questionnaires have 30 developmental items divided into five domains: Communication, Gross Motor, Fine Motor, Problem Solving, and Personal-social development. The questionnaire must be completed by the parent or the caregiver of the child. Answers to items are rated *yes*, *sometimes,* or *not yet*, and are scored as 10, 5, or 0, respectively. Each questionnaire is administered to children of that age and in the ±1-month range. Domain cut-off scores were calculated on a 2 SD basis. It was suggested that children whose questionnaire results stood at, or below, the established cut-off score in one or more of the domains should be referred for further assessment. The questionnaires were translated from English into Italian by two native Italian speaker neuropsychiatrists (FM and CS) and then translated back into English by a blinded translator who was not directly involved in the study.

### 2.5. Griffiths Scales of Child Development, Third Edition (Griffiths III)

Griffiths III is a comprehensive, child-friendly developmental measure for continuous use from birth (one month) to five years and 11 months (71 months). Griffiths III will provide an overall measure of a child’s development, as well as an individual profile of strengths and needs across five areas: (1) Foundations of Learning—assessing critical aspects of learning during early childhood years; (2) Language and Communication—measuring overall language development, including expressive language, receptive language and (to a lesser extent) use of language to communicate socially with others. (3) Eye and Hand Coordination—considering fine motor skills, manual dexterity, and visual perception skills. (4) Personal-Social-Emotional—measuring constructs related to the child’s developing sense of self and growing independence, interactions with others, and many aspects of emotional development. (5) Gross Motor—assesses postural control, balance, and gross body coordination, among other abilities. Griffiths III provides raw scores, standard scores, age equivalents, sub-quotients, general quotients, and percentile equivalents. Italian adaptation of the instrument showed good reliability: for the total sample, KR-20 indices ranged from 0.98 to 0.99, whereas for the age-specific subgroups, these indices ranged from 0.83 to 0.98 [32].

### 2.6. Peabody Developmental Motor Scale-Second Edition

The Italian version of the Peabody Developmental Motor Scale, Second Edition (PDMS-2) is an object-based performance test providing for a multidimensional assessment of fine and gross motor abilities in preschoolers (from birth to six years). The gross-motor quotient (QGM), the fine motor quotient (QFM), and the total-motor quotient (QMT) are three global indicators of motor performance derived from this set of six subtests (249 items total). Items of the PDMS-2 are scored on a three-point scale (0, 1, and 2). A score of 2 is assigned when the child performs the item according to the specified item criterion; a score of one indicates that the behavior is emerging but the criterion for successful performance is not fully met; finally, a score of zero indicates that the child cannot or will not attempt the item or that the attempt does not show that the skill is emerging. Developmental quotients for the GM, FM, and TM composites were then derived by summing the subtest standard scores and converting them to a quotient with a mean of 100 and a standard deviation of 15. All scores showed very good reliability: in the Italian validation sample, Cronbach’s α ranged from 0.90 to 0.99; moreover, in the two gender-specific subsamples, Cronbach’s α ranged from 0.87 to 0.99 [33].

### 2.7. The Developmental Profile—Third Edition

The Developmental Profile, Third Edition (DP-3) is a norm-referenced developmental screening instrument that is administered as a structured parent interview to determine the child’s present level of functioning; administration as a parent/caregiver checklist is an option when a direct interview is not feasible. It is designed for use to assess children from birth through to 12 years and 11 months. The DP-3 inventories 180 skills and is designed to assess a child’s development on five scales: Physical, Adaptive Behavior, Social-Emotional, Cognitive, and Communication. A General Development Score may also be obtained when all five scales are administered. DP-3 scores are available in five formats: standard scores, percentile ranks, stanines, age equivalents, and descriptive ranges. Italian adaptation showed adequate reliability: in the 13 age-specific subgroups the split half coefficients ranged from 0.64 to 0.99; in detail, only six coefficients, equal to 7.70%, were lower than 0.70 [34].

### 2.8. Data analysis

All statistical analyses were carried out using IBM SPSS Statistics version 23.0 (SPSS Inc., Chicago, IL, USA), and MPLUS 3.0 [35].

Correlation analyses were estimated by Pearson’s correlation coefficient. To evaluate item discrimination power, corrected item-total correlations have been computed. To evaluate reliability, Cronbach’s alfa indices and test-retest reliability, using paired sample t-tests, have been conducted.

The dimensionality of the test was tested with a Confirmative Factor Analysis (CFA) for dichotomous data using Mplus 3.0 software that implemented the Weighted Least Squares Means and Variance adjusted (WLSMV) estimation method [36]. WLSMV uses weighted least square parameter estimates from the diagonal of the weight matrix. This method is recommended for categorical variables [35,36,37] on the basis of simulation studies [38]. In order to evaluate the fit of the model, the chi-square, the Comparative Fit Index (CFI) [39], the Tucker-Lewis Index (TLI) [40], and the Root Mean Square Error of Approximation (RMSEA) statistics were used [41]. Bentler and Bonnet [39,42] suggested that values higher than 0.90 for the CFI and TLI indicate that the model provides an adequate fit to the data and Browne and Cudeck [42] suggested that values of RMSEA lower than 0.05 indicate a close fit.

The ASQ-3 validity was analyzed by comparing the performances on the questionnaires completed by the group with typical development and the clinical group.

Simultaneous validity was measured by comparing children’s performance classification against a standardized test and their performance classification at ASQ-3. The agreement meant that the ASQ-3 had assigned a child to the same classification as the standardized test had; disagreement meant that the ASQ-3 classification did not match the standardized test’s classification. The identified clinical group was recruited from the Department of Human Neuroscience and the Griffiths Scales of Child Development, Third Edition, Peabody Developmental Motor Scale, Second Edition, and the Developmental Profile, Third Edition tools were used.

We generated receiver operating characteristic curves (ROC) for the ASQ-3 scores. Youden’s index was calculated to identify the optimal cut-off score that maximizes sensitivity and specificity for each domain; moreover, a diagnostic cutoff has been identified using the criterion of specificity equal to 0.99. The scores of the group with typical development and those of the clinical group were combined, as it seemed to be the most appropriate procedure for screening programs to select scores maximizing accuracy and minimizing error.

Finally, clinical validity was measured using comparative analyses between mean domain scores for children with typical development and clinical groups. A p-value <0.05 represented statistical significance for all tests.

## 3. Results

Data were collected from a total of 2278 Italian children (M = 1238; F = 1040, range 1–66 months). 1874 children (M = 987; F = 887) were recruited from 127 kindergartens located in Italy, and the sample was divided into 21 age stages. The clinical group was composed of 404 children (M = 251; F = 153), of whom 50 completed the concurrent assessment. Mothers accounted for 88% of persons who completed the questionnaires. Table 1 summarizes demographic characteristics.

The item’s significant discriminative capacity emerged from the analysis of the Italian sample. As shown in Table 2, the corrected item-total correlation was calculated and found to be high in all five ASQ-3 developmental areas (ranging from 0.44 to 0.98). All correlations were significant at *p* < 0.001. The reliability of the questionnaires has been studied by examining their internal consistency and test–retest reliability.

Cronbach’s alpha value was calculated for developmental area scores for 21 age intervals (see Table 3). These alphas indicate that ASQ items have good-to-acceptable internal consistency.

Test–retest reliability was analyzed by comparing ASQ-3 questionnaires completed by the same parent (*n* = 50), within a two weeks’ interval. Pearson’s correlations ranged from 0.82 to 0.94, indicating strong test–retest reliability across ASQ developmental areas. The differences between the mean scores of the two evaluations were not statistically significant (Table 4).

The CFA demonstrated a satisfactory fit model [*X*^2^ (395) = 2343.78, *p* < 0.001, CFI = 0.98, TLI = 0.98, RMSEA (90%C.I.) = 0.047, (0.045–0.048)] attested the construct validity of the Italian adaptation. As shown in Table 5, all factor loadings were statistically significant (*p* < 0.001).

Table 6 shows the correlations between ASQ-3 developmental areas and overall scores ranging from 0.73 to 0.88. Correlations between ASQ-3 domains were also calculated, showing the lowest correlation between gross motor and communication domain (*r* =0.73) and the highest, between problem-solving and motor fine domain (*r* = 0.88). All correlations were positive and significant at *p* < 0.001.

The comparison group between children with typical development (TD) and children with NDs (i.e., autism spectrum disorder, language disorders, motor delay, global developmental delay) are summarized in Table 7, Table 8, Table 9 and Table 10. With regard to concurrent validity, ASQ-3 domains were significantly correlated with GDMS-3, PDMS-2, and DP-3 subscales. This means that the response pattern on the ASQ-3 correlates with the response pattern on a similar measure of child development. For the ASQ-3, TD and clinical groups were combined for all analyses and determination of the age interval cut-off scores.

Table 11 showed the two cut-off values identified by the ROC analyses: the Youden’s index has been used as a screening index, whereas a diagnostic index has been identified using the criterion of specificity = 0.99.

According to the diagnostic index, a cut-off > −1.41 z-score discriminated best between patients with and without NDs, for the fine-motor scale.

## 4. Discussion

The present study investigated the psychometric characteristics of the Italian version of the ASQ-3, a screening tool for NDs, in a sample of 2278 children attending kindergarten, who participated in the Italian adaptation of the instrument. Although it was easily administered in a school context and in a clinical setting, and valued by parents for developmental screening, it was culturally sensitive. Similar to other studies, the answers to some questions were deemed culturally sensitive and were therefore adjusted to make the questionnaire more culturally relevant [19,20,22,23,43,44].

In addition to a general societal consensus that early detection of NDs was a worthwhile goal, ASQ-3 was translated into several languages and has demonstrated its adaptability in diverse cultural environments [43,44,45].

The data yielded by the present study confirmed that the school system actively responded to the project, and a large proportion of the parents participated in the study, completing the questionnaire and returning it to the school. Only 1% of the questionnaires (21 out of 2299) could not be used as they had been completed incorrectly. This picture is in keeping with those reported by other screening programs in the literature [46]. Since participation in screening programs was voluntary, a certain amount of non-adherence was to be expected [47]. In our study, all parents adhered to the screening program. Empowering interventions and programs had a positive impact on the mental health of parents, especially mothers.

In addition, the low proportion of incorrectly completed questionnaires indirectly confirmed that the ASQ-3 was relatively easy to complete and allowed parents to conduct a guided observation of their children’s developmental age, by investigating the children’s skills in daily life.

As regards reliability, results showed evidence for adequate internal consistency reliability, and a high test-retest reliability attesting the reliability of the Italian adaptation of the test. Analyses of fit indices and factor loadings supported the five-factor solution for all questionnaires as found in previous studies [26].

In line with the ASQ-3 original study [28] and others [26,48] significant correlations were computed between the total domain score and the overall score for each questionnaire. The strong correlation coefficients between the total score and the subscale score support the validity of the ASQ-3. However, several studies have reported great variation in scores, conflicting sensitivity, and predictive value of ASQ scores for different age questionnaire intervals [49,50,51].

As regards validity, results showed positive, large domain-total correlations, as well as moderate to large correlations between developmental domains across questionnaire age ranges. This implies that items can measure the same general concept of child development. Matched items and corresponding domain scores from the ASQ-3 and concurrent assessment tools (GDMS-3, PDMS-2, DP-3) also supported this implication. This means that ASQ-3 can be used with confidence to accurately detect children who are likely to suffer from developmental difficulties.

Malak et al. [52] highlighted that ASQ-3 is a valid and comparable instrument to assessments made by specialists. This information might help parents feel more empowered when interacting with their children.

One aspect deserving consideration regards the averages and the standard deviation of the US and Italian samples. A substantial overlap emerged between the average values in the age groups considered, although values in the Italian sample are always higher than those in the US sample. Italian values are actually closer to those yielded by other validations of the ASQ-3 than to the American values [53]. As concerns the SD, however, Italian values appear to be qualitatively lower than those of the US. This acquires much more significance since the cut-off for being defined either “at risk” or “clinical” is based on the rule of the mean minus 1 or 2 SD; the cut-off point used to define the risk and clinical areas in the selected sample is consequently strongly dependent on these values. According to the formula used to define the cut-off value, if a lower SD were to persist even when the size of the Italian sample increases, then the authors could assume that the Italian cut-off values need to be different to the US cut-off values (see Table 11).

It is important to note that the optimal cut-off scores (Youden Index) in the present study provide the best balance of sensitivity and specificity for identifying a disorder in Italian children.

As regards the clinical groups with respect to the TD group, parents of children with ASD or GDD fell within the clinical range in all ASQ-3 developmental areas (Table 7 and Table 9). When we considered the group with language disorder and compared it with the TD group we identified an overlapping score in the Personal-social domain (Table 8). It is possible that children are perceived by their parents as being less skillful on the social than on the language level. When we consider the group with a motor delay with respect to compare with the TD group we identified a more significant difference in the raw score in the Gross and Fine motor domains (Table 10). These results confirm the appropriateness of the ASQ-3 tool in clinical practice.

## 5. Conclusions

To conclude, ASQ-3 scores may help identify risk in children with mild-to-severe neurodevelopmental disorders. Our results indicate that current methods of caregiver reporting for early childhood development may be optimal. Children screening positive for the developmental delay must be referred for a more thorough assessment by an early child development specialist to validate the screen and establish what type of intervention is the most appropriate. Early identification and accurate assessment are important starting points to better understand and anticipate the children’s needs and links to services. It was observed that the ASQ-3 met the psychometric properties necessary for a targeted and systematic assessment of development during health daycare with cutoff scores adapted to the local 1–66-month-old population. Findings in this study suggest that developmental screening tools may offer valuable information to all professionals who work with the children such as neuropsychiatrists, pediatricians, clinical psychologists, occupational therapists, physical therapists, and speech therapists.

## Figures and Tables

**Table 1 ijerph-20-05014-t001:** Characteristics of the study participants.

ASQ-3 Age Interval (Months)	*n*	Sex				Italy					
		M		F		Northern		Center		Southern	
		f	%	f	%	f	%	f	%	f	%
2	97	51	52.6	46	47.4	30	30.9	59	60.8	8	8.2
4	100	58	58.0	42	42.0	41	41.0	50	50.0	9	9.0
6	86	52	50.5	34	39.5	18	20.9	44	51.2	24	27.9
8	96	51	53.1	45	46.9	41	42.7	43	44.8	12	12.5
9	92	46	50.0	46	50.0	34	37.0	42	45.7	16	17.4
10	94	51	54.3	43	45.7	40	42.6	25	26.6	29	30.9
12	107	56	52.3	51	47.7	47	43.9	52	48.6	8	7.5
14	95	50	52.6	45	47.4	25	26.3	55	57.9	15	1.8
16	93	46	49.5	47	50.5	28	30.1	56	60.2	9	9.7
18	137	79	57.7	58	42.3	54	39.4	55	40.1	28	20.4
20	105	56	53.3	49	46.7	45	42.9	45	42.9	15	14.3
22	99	55	55.6	44	44.4	38	38.4	40	40.4	21	21.2
24	86	49	57.0	37	43.0	30	34.9	46	53.5	10	11.6
27	98	52	53.1	46	46.9	34	34.7	37	37.8	27	27.6
30	86	46	53.5	40	46.5	35	40.7	38	44.2	13	15.1
33	90	47	52.2	43	47.8	33	36.7	53	58.9	4	4.4
36	148	78	52.7	70	47.3	29	19.6	109	73.6	10	6.8
42	166	92	55.4	74	44.6	44	26.5	103	62.0	19	11.4
48	165	87	52.7	78	47.3	40	24.2	119	72.1	6	3.6
54	112	62	55.4	50	44.6	27	24.1	68	60.7	17	15.2
60	126	74	58.7	52	41.3	33	26.2	75	59.5	18	14.3
TOTAL	2278	1238	54.3	1040	45.7	746	32.7	1214	53.3	318	14.0

**Table 2 ijerph-20-05014-t002:** Corrected item-total correlations of ASQ-3 developmental areas.

ASQ-3 Age Interval (Months)	Communication	Gross Motor	Fine Motor	Problem Solving	Personal-Social
2	0.72	0.69	0.73	0.81	0.72
4	0.73	0.74	0.74	0.88	0.68
6	0.64	0.71	0.75	0.76	0.79
8	0.85	0.78	0.62	0.75	0.50
9	0.92	0.96	0.89	0.98	0.89
10	0.87	0.86	0.92	0.91	0.80
12	0.70	0.88	0.80	0.78	0.84
14	0.77	0.67	0.76	0.78	0.85
16	0.52	0.32	0.53	0.74	0.44
18	0.73	0.75	0.72	0.75	0.80
20	0.73	0.48	0.52	0.38	0.54
22	0.53	0.63	0.68	0.75	0.83
24	0.72	0.71	0.58	0.60	0.67
27	0.88	0.86	0.86	0.80	0.79
30	0.88	0.66	0.68	0.78	0.64
33	0.81	0.60	0.64	0.56	0.58
36	0.77	0.59	0.64	0.52	0.53
42	0.64	0.48	0.60	0.52	0.48
48	0.69	0.47	0.56	0.50	0.48
54	0.80	0.54	0.69	0.65	0.66
60	0.86	0.65	0.76	0.72	0.72

Legend: All correlations were significant at *p* < 0.001.

**Table 3 ijerph-20-05014-t003:** Cronbach’s alpha coefficient of ASQ-3 developmental areas.

ASQ-3 Age Interval (Months)	Communication	Gross Motor	Fine Motor	Problem Solving	Personal-Social
2	0.90	0.88	0.89	0.94	0.89
4	0.93	0.89	0.90	0.96	0.87
6	0.83	0.88	0.88	0.90	0.92
8	0.94	0.91	0.82	0.88	0.74
9	0.97	0.94	0.96	0.98	0.95
10	0.94	0.95	0.93	0.92	0.92
12	0.87	0.96	0.92	0.92	0.94
14	0.91	0.85	0.89	0.90	0.95
16	0.76	0.53	0.76	0.88	0.65
18	0.88	0.91	0.86	0.89	0.72
20	0.89	0.74	0.72	0.61	0.76
22	0.73	0.80	0.84	0.88	0.94
24	0.88	0.86	0.80	0.83	0.86
27	0.96	0.94	0.94	0.93	0.91
30	0.96	0.83	0.86	0.91	0.83
33	0.93	0.82	0.85	0.79	0.81
36	0.92	0.82	0.84	0.76	0.75
42	0.84	0.74	0.82	0.76	0.72
48	0.87	0.73	0.80	0.74	0.71
54	0.92	0.76	0.87	0.84	0.84
60	0.95	0.85	0.91	0.89	0.89

**Table 4 ijerph-20-05014-t004:** Mean and standard deviation (raw scores) and test-retest correlation of ASQ-3 developmental areas.

		First Evaluation (*n* = 50)		Second Evaluation (*n* = 50)		
	*r*	M	SD	M	SD	Mean Absolute Difference
Communication	0.90	55.00	6.85	56.70	4.80	1.70
Gross Motor	0.94	55.60	7.40	57.20	5.07	1.60
Fine Motor	0.88	53.40	7.32	55.50	6.57	2.10
Problem Solving	0.84	53.80	8.30	56.10	5.83	2.30
Personal-Social	0.82	53.90	6.65	56.40	5.35	2.50

**Table 5 ijerph-20-05014-t005:** Factors loading of ASQ-3.

Item	Communication	Gross Motor	Fine Motor	Problem Solving	Personal-Social
1	0.86	0.80	0.76	0.83	0.80
2	0.91	0.84	0.85	0.81	0.80
3	0.94	0.91	0.84	0.89	0.86
4	0.90	0.86	0.88	0.91	0.82
5	0.89	0.88	0.89	0.88	0.86
6	0.89	0.90	0.86	0.89	0.84

Legend: All data were significant at *p* < 0.001.

**Table 6 ijerph-20-05014-t006:** Correlations between ASQ-3 developmental area scores.

			Area		
	Communication	Gross Motor	Fine Motor	Problem Solving	Personal-Social
Communication	-				
Gross Motor	0.73	-			
Fine Motor	0.79	0.84	-		
Problem Solving	0.85	0.83	0.88	-	
Personal-Social	0.82	0.83	0.84	0.85	-

Legend: All correlations were significant at *p* < 0.001.

**Table 7 ijerph-20-05014-t007:** Comparison group between children with Autism Spectrum Disorder and Typical Development on ASQ-3.

	Typical Development (*n* = 1874)		Autism Spectrum Disorder (*n* = 51)		*t*	*df*	*p*	Cohen’s d
	M	SD	M	SD				
Communication	56.22	6.99	20.90	17.28	33.16	1922	<0.001	1.51
Gross Motor	55.34	8.47	30.50	9.33	20.43	1922	<0.001	0.93
Fine Motor	54.48	8.99	20.40	12.53	26.14	1922	<0.001	1.19
Problem Solving	55.06	8.65	21.70	13.69	26.42	1922	<0.001	1.21
Personal-Social	55.06	7.34	29.40	12.44	23.82	1922	<0.001	1.09

Legend: d ≤ 0.20: small, 0.21 ≤ d ≤ 0.50: moderate, 0.51 ≤ d ≤ 0.80: medium, d ≥ 0.81: large.

**Table 8 ijerph-20-05014-t008:** Comparison group between children with Language Disorder and Typical Development on ASQ-3.

	Typical Development (*n* = 1874)		Language Disorder (*n* = 51)		*t*	*df*	*p*	Cohen’s d
	M	SD	M	SD				
Communication	56.22	6.99	35.60	10.28	20.29	1922	<0.001	0.93
Gross Motor	55.34	8.47	50.70	6.47	3.85	1922	<0.001	0.18
Fine Motor	54.48	8.99	46.70	8.12	6.05	1922	<0.001	0.28
Problem Solving	55.06	8.65	46.20	8.36	7.16	1922	<0.001	0.33
Personal-Social	55.06	7.34	55.10	7.73	-0.04	1922	0.97	-

Legend: d ≤ 0.20: small, 0.21 ≤ d ≤ 0.50: moderate, 0.51 ≤ d ≤ 0.80: medium, d ≥ 0.81: large.

**Table 9 ijerph-20-05014-t009:** Comparison group between children with Global Developmental Delay and Typical Development on ASQ-3.

	Typical Development (*n* = 1874)		Global Developmental Delay (*n* = 51)		*t*	*df*	*p*	Cohen’s d
	M	SD	M	SD				
Communication	56.22	6.99	10.50	11.66	44.64	1922	<0.001	2.03
Gross Motor	55.34	8.47	23.50	11.17	26.01	1922	<0.001	1.19
Fine Motor	54.48	8.99	11.40	9.95	33.35	1922	<0.001	1.52
Problem Solving	55.06	8.65	12.90	10.36	33.84	1922	<0.001	1.54
Personal-Social	55.06	7.34	16.30	12.11	36.05	1922	<0.001	1.64

Legend: d ≤ 0.20: small, 0.21 ≤ d ≤ 0.50: moderate, 0.51 ≤ d ≤ 0.80: medium, d ≥ 0.81: large.

**Table 10 ijerph-20-05014-t010:** Comparison group between children with Motor Delay and Typical Development on ASQ-3.

	Typical Development (*n* = 1874)		Motor Delay (*n* = 51)		*t*	*df*	*p*	Cohen’s d
	M	SD	M	SD				
Communication	56.22	6.99	46.70	16.59	8.99	1922	<0.001	0.41
Gross Motor	55.34	8.47	17.60	8.88	31.08	1922	<0.001	1.42
Fine Motor	54.48	8.99	23.60	12.66	23.68	1922	<0.001	1.08
Problem Solving	55.06	8.65	33.80	13.83	16.83	1922	<0.001	0.77
Personal-Social	55.06	7.34	34.60	14.24	18.79	1922	<0.001	0.86

Legend: d ≤ 0.20: small, 0.21 ≤ d ≤ 0.50: moderate, 0.51 ≤ d ≤ 0.80: medium, d ≥ 0.81: large.

**Table 11 ijerph-20-05014-t011:** Cut-off the Italian version of ASQ-3 (z-score).

		Youden Index (Screening)	Diagnostic Index
Communication	Cutoff	−0.65	−1.42
	Sensibility	0.85	0.67
	Specificity	0.97	0.99
Gross motor	Cutoff	−0.50	−1.57
	Sensibility	0.86	0.57
	Specificity	0.90	0.99
Fine motor	Cutoff	−0.48	−1.41
	Sensibility	0.86	0.70
	Specificity	0.95	0.99
Problem solving	Cutoff	−0.35	−1.57
	Sensibility	0.91	0.59
	Specificity	0.90	0.99
Personal-social	Cutoff	−0.60	−1.47
	Sensibility	0.83	0.59
	Specificity	0.94	0.99

## Data Availability

The data presented in this study are available on request from the corresponding author.

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
