# Peer review of "Psychometric Properties and Validation of the Italian Version of Ages & Stages Questionnaires Third Edition"

_ijerph, 2023, doi:10.3390/ijerph20065014_

Round 1

Reviewer 1 Report

Dear Authors,

Thank you for the opportunity to review an interesting article entitled: ‘Psychometric properties and validation of the Italian version of Ages & Stages Questionnaires third edition’. The aim of this study is to determine a reliable and valid instrument for the Italian population to enable the screening of children's development aged 1 to 66 months. This topic is of a great importance due to the increasing prevalence of various developmental disorders in children. This problem is also indicated by the nomenclature changes introduced in the ICD-11 in the category of neurodevelopmental disorders.

The strengths of the article presented for evaluation are the adaptation and validation of a commonly used tool to Italian conditions, as well as the robust empirical research carried out on a large sample, the use of appropriate research and analytical procedures and the proper conduct of statistical analyses.

The reviewer's job, on the other hand, is to help improve the article so that it meets the highest possible standards of the journal, therefore I will focus on its weaknesses.

Introduction

[1].  Missing from the introduction is information on ND scales in countries other than the USA. And since the ASQ-3 is 'widely used' (as the authors point out - lines 64-65), it is worth supplementing the text with results from other countries.

[2].  The adapted questionnaire should already be described in detail in the introduction. It is useful to outline the procedure of its development together with the results of psychometric analyses of the original questionnaire.

Methods

[3].  The article lacks the subsection 'Adaptation procedure', which should have included information on how the Italian adaptation of the questionnaire was developed in general.

[4].  I suggest changing the order of the 'Methods' section. It would be good to describe the adaptation procedure first, then the study, followed by the study participants, the analytical procedures and the tools.

[5].  The description of the tools lacks information on how each item was assessed and the value of the Cronbach's α reliability coefficient of the tools used. Has Cronbach's α been checked on your own sample for the tools used? If not, it needs to be added and credited.

Results

[6].  It is worth considering changing Cronbach's α to McDonald's ω reliability coefficient or reporting both results.

[7].  The notation of the designations 'p', 't', 'df', etc., which should be in italics each time, should be standardised.

References

[8].  DOI numbers should be completed in the bibliography.

Author Response

Reviewer #1

Dear Authors,

Thank you for the opportunity to review an interesting article entitled: ‘Psychometric properties and validation of the Italian version of Ages & Stages Questionnaires third edition’. The aim of this study is to determine a reliable and valid instrument for the Italian population to enable the screening of children's development aged 1 to 66 months. This topic is of a great importance due to the increasing prevalence of various developmental disorders in children. This problem is also indicated by the nomenclature changes introduced in the ICD-11 in the category of neurodevelopmental disorders.

The strengths of the article presented for evaluation are the adaptation and validation of a commonly used tool to Italian conditions, as well as the robust empirical research carried out on a large sample, the use of appropriate research and analytical procedures and the proper conduct of statistical analyses.

The reviewer's job, on the other hand, is to help improve the article so that it meets the highest possible standards of the journal, therefore I will focus on its weaknesses.

Introduction

[1].  Missing from the introduction is information on ND scales in countries other than the USA. And since the ASQ-3 is 'widely used' (as the authors point out - lines 64-65), it is worth supplementing the text with results from other countries.

Authors’ reaction: We would like to thank the reviewer for the suggestion. A new information has been added in the manuscript (from line 75 to line 80).

[2].  The adapted questionnaire should already be described in detail in the introduction. It is useful to outline the procedure of its development together with the results of psychometric analyses of the original questionnaire.

Authors’ reaction: The manuscript was revised accordingly. A new information has been added in the manuscript (from line 81 to line 85).

Methods

[3].  The article lacks the subsection 'Adaptation procedure', which should have included information on how the Italian adaptation of the questionnaire was developed in general.

Authors’ reaction: The subsection “Adaptation procedure” has been added in the manuscript (see line 93).

[4].  I suggest changing the order of the 'Methods' section. It would be good to describe the adaptation procedure first, then the study, followed by the study participants, the analytical procedures and the tools.

Authors’ reaction: The manuscript was revised.

[5].  The description of the tools lacks information on how each item was assessed and the value of the Cronbach's α reliability coefficient of the tools used. Has Cronbach's α been checked on your own sample for the tools used? If not, it needs to be added and credited.

Authors’ reaction: The value of the Cronbach's α reliability coefficient of the tools used has been added in the manuscript (see page 4, from line 163 to line 166; from line 180 to line 182; from line 190 to line 193).

Results

[6].  It is worth considering changing Cronbach's α to McDonald's ω reliability coefficient or reporting both results.

Authors’ reaction: we didn’t calculate McDonald's ω reliability coefficient because the data were not normally distributed. According to Trizano-Hermosilla and Alvarado (2016) who recommended using ω whenever items have a normal distribution, we didn’t compute this reliability coefficient (Trizano-Hermosilla, I.  & Alvarado, J.M. Best alternatives to Cronbach’s alpha reliability in realistic conditions: Congeneric and asymmetrical measurements. Frontiers in Psychology, 2016, 7: 769).

[7].  The notation of the designations 'p', 't', 'df', etc., which should be in italics each time, should be standardised.

Authors’ reaction: The manuscript was revised. The designations 'p', 't', 'df', etc., has been modified.

References

[8].  DOI numbers should be completed in the bibliography.

Authors’ reaction: The manuscript was revised. The DOI numbers has been added to the bibliography.

Reviewer 2 Report

First of all, I would like to congratulate the authors for this much-needed work. Early detection of developmental delays, in any of the developmental areas, is important to provide early appropriate specialty intervention so that the impact on both the child and their families can be reduced. The ASQ-3 is a tool widely used by many professionals around the world.

I consider that the work is adequate, although I would recommend that the authors specify some points of the procedure (how many nursery schools participated, what was the eligible sample, was the sample size initially calculated, was the information collected in paper or digital format). , if it was collected in paper format or digitally) advocating for the transparency and reproducibility of science.

In addition, I encourage authors to be explicit and mention in their article all professionals who work with children with developmental delays other than pediatricians, such as occupational therapists, physical therapists, and speech therapists. All of them are likely to use the ASQ-3 in their assessments.

For the rest, I consider that the manuscript is of sufficient quality and interest for its publication in IJERPH.

Author Response

Reviewer #2

First of all, I would like to congratulate the authors for this much-needed work. Early detection of developmental delays, in any of the developmental areas, is important to provide early appropriate specialty intervention so that the impact on both the child and their families can be reduced. The ASQ-3 is a tool widely used by many professionals around the world.

I consider that the work is adequate, although I would recommend that the authors specify some points of the procedure (how many nursery schools participated, what was the eligible sample, was the sample size initially calculated, was the information collected in paper or digital format), if it was collected in paper format or digitally) advocating for the transparency and reproducibility of science.

Authors’ reaction:  A new information has been added in the manuscript (see page 3 line 100, and page 5 line 236).

was the sample size initially calculated

The purpose of this study was to collect 1.680 children in order to have 80 children in each age range. In the original version of the test, there were 21 age ranges: we decided to maintain this division; for that reason, we decided to collect around 80 children for each age range in order to have a representative sample of the Italian population, an adequate sample size to calculate standard scores in each age range and to perform statistical analyses according to the original version of the test.

In addition, I encourage authors to be explicit and mention in their article all professionals who work with children with developmental delays other than pediatricians, such as occupational therapists, physical therapists, and speech therapists. All of them are likely to use the ASQ-3 in their assessments.

Authors’ reaction: A new information has been added in the manuscript (see page 11, from line 381 to line 382)

For the rest, I consider that the manuscript is of sufficient quality and interest for its publication in IJERPH.

Round 2

Reviewer 1 Report

Comments and Suggestions for Authors:

1. The notation of the designations 'p', 't', 'df', 'r', etc., which should be in italics each time, should be standardised.
2. In Line 264 should be χ2 not χ2.

3. In the bibliography, the year of publication should be in bold.

Author Response

Point 1. The notation of the designations 'p', 't', 'df', 'r', etc., which should be in italics each time, should be standardised.

We would like to thank the reviewer for the suggestion. The manuscript was revised according to the request.

Point 2. In Line 264 should be χ2 not χ2.

We changed χ2 to χ2.

Point 3. In the bibliography, the year of publication should be in bold.

The content was revised according to the request.